# Large language models implicitly learn to straighten neural sentence trajectories to construct a predictive representation of natural language

**Eghbal A. Hosseini**
Brain and Cognitive Sciences, McGovern Institute for Brain Research
Massachusetts Institute of Technology
Cambridge, MA, 02139
ehoseini@mit.edu

**Evelina Fedorenko**
Brain and Cognitive Sciences, McGovern Institute for Brain Research
Massachusetts Institute of Technology
Cambridge, MA, 02139
evelina9@mit.edu

## Abstract

Predicting upcoming events is critical to our ability to effectively interact with our environment and conspecifics. In natural language processing, transformer models, which are trained on next-word prediction, appear to construct a general-purpose representation of language that can support diverse downstream tasks. However, we still lack an understanding of how a predictive objective shapes such representations. Inspired by recent work in vision neuroscience Hénaff et al. (2019), here we test a hypothesis about predictive representations of autoregressive transformer models. In particular, we test whether the neural trajectory of a sequence of words in a sentence becomes progressively more straight as it passes through the layers of the network. The key insight behind this hypothesis is that straighter trajectories should facilitate prediction via linear extrapolation. We quantify straightness using a 1-dimensional curvature metric, and present four findings in support of the trajectory straightening hypothesis: i) In trained models, the curvature progressively decreases from the first to the middle layers of the network. ii) Models that perform better on the next-word prediction objective, including larger models and models trained on larger datasets, exhibit greater decreases in curvature, suggesting that this improved ability to straighten sentence neural trajectories may be the underlying driver of better language modeling performance. iii) Given the same linguistic context, the sequences that are generated by the model have lower curvature than the ground truth (the actual continuations observed in a language corpus), suggesting that the model favors straighter trajectories for making predictions. iv) A consistent relationship holds between the average curvature and the average surprisal of sentences in the middle layers of models, such that sentences with straighter neural trajectories also have lower surprisal. Importantly, untrained models don't exhibit these behaviors. In tandem, these results support the trajectory straightening hypothesis and provide a possible mechanism for how the geometry of the internal representations of autoregressive models supports next word prediction.

37th Conference on Neural Information Processing Systems (NeurIPS 2023).

# 1 Introduction

Biological systems, like brains, and artificial systems, like deep neural networks, construct internal representations in the service of their internal or external objectives. Certain objectives appear to yield representations that are useful across diverse behaviors. For example, representations that are predictive of incoming input have been argued to be useful in both biological systems - across perception, action, and cognition (Rao and Ballard, 1999; Palmer et al., 2015; Shadmehr et al., 2010; Hohwy et al., 2008; Jessup et al., 2010; Shain et al., 2020; Frank et al., 2015) - and in artificial systems across domains (van den Oord et al., 2018; Radford et al., 2018). Two general approaches have been commonly used in modeling predictive processing in the brain. The first approach leverages information theory (Shannon, 1949) to quantify the relationship between the past and current neural states and future inputs (e.g., Bialek et al., 2007; Tishby et al., 2000; Wiskott and Sejnowski, 2002; Palmer et al., 2015). A key limitation of this kind of an approach is that they do not specify how the information about past inputs is actually used to make predictions (see Hénaff, 2018 for discussion). The second approach instead focuses on circuit-level motifs—specifically interactions between lower-level and higher-level areas— that give rise to a predictive, top-down signal, and the bottom-up error signal (e.g., Rao and Ballard, 1999). This approach faces the challenge of specifying what information is represented in high-level cortical areas, which, for many domains, remains not well understood.

Recently, in the context of visual processing, Henaff (2018; see also Hénaff et al., 2019) have developed an approach to temporal prediction at an intermediate level of abstraction. In contrast to the information-theory-grounded approaches, which focus on predicting upcoming inputs Palmer et al., 2015, this approach focuses on the internal representation states and on predicting future internal states. The critical insight comes from vision: because a sequence of visual inputs to the retina evolves in a nonlinear manner, and are difficult to extrapolate, visual system performs a series of transformation to make them easier to predict. The representation of input sequence is transformed to result in **straighter** the trajectory in the internal state, and allow for linear extrapolation of future states of the sequence. Hénaff et al. (2019) found support for this straightening hypothesis in behavioral psychophysics experiments and in neural recordings in the early visual areas of macaques (Hénaff et al., 2021). They also tested the predictions of this hypothesis in AlexNet (Krizhevsky et al., 2012), an early convolutional neural network for vision, but did not observe representation straightening. They hypothesized that representation straightening may only emerge in systems where the objective has to do with prediction (cf. AlexNet where the objective function is object categorization).

Language is a domain where information unfolds over time and where prediction is a natural objective function. Indeed, many successful language models use next-word prediction as their core training objective (e.g., Radford et al., 2018). As a result, representational straightening seems a priori plausible as a mechanism for linguistic prediction. Here, we evaluate the straightening hypothesis across four computational experiments. In Experiment 1, we show that across a corpus of approximately 8.5K human-generated sentences, the average curvature of sentences decreases gradually between the input layer and the deep layers. In Experiment 2, we show that larger models and models that are trained on larger datasets show a greater degree of representation straightening, suggesting that straighter internal representations is what allows for better predictive performance. In Experiment 3, we perform a critical comparison between natural, human-generated sentences and model-generated sentences (created by providing the models with the first few words of the natural sentences) and show that the model-generated sentences have straighter trajectories. Finally, in Experiment 4, we relate sentence curvature to surprisal, a measure of how expected words are in context (Shannon, 1949), which has been shown to predict human behavior and neural responses to language (Levy, 2008; Smith and Levy, 2013; Willems et al., 2016; Henderson et al., 2016; Lopopolo et al., 2017; Shain et al., 2020; Heilbron et al., 2022). Jointly, these results provide evidence for representation straightening as a mechanistic hypothesis about the computations that allow transformer language models to construct a predictive representation in the service of their behavioral objective.

## 2 Methods

### 2.1 Models

We focused on autoregressive language models (Brown et al., 2020; Radford et al., 2018), specifically the GPT model family (**Figure 1A**). These models are explicitly trained on the next-word prediction objective. Each layer consists of 4 main blocks: (i) 1st layer normalization, (ii) self-attention, (iii) 2nd layer normalization, and (iv) the feed-forward layer. For most analyses, we used GPT2-XL (48 layers, 1600 embedding dimensions, 25 heads, 1558M parameters). To examine the effects on representational geometry of model training, we used GPT2 (12 layers, 768 embedding dimensions, 12 heads, 117M parameters; Radford et al., 2018); and to examine the effects of model size, we additionally used GPT2-Large (36 layers, 1280 embedding dimensions, 20 heads, 774M parameters; Radford et al., 2018), GPT2-Medium (24 layers, 1024 embedding dimensions, 16 heads, 345M parameters; Radford et al., 2018), and distillGPT2 (6 layers, 768 embedding dimensions, 12 heads, 82M parameters; Sanh et al., 2019). For all analyses, we used pre-trained models from the HuggingFace library (Wolf et al., 2019).

### 2.2 Curvature estimation

We developed a curvature metric using the neural trajectory of words (tokens) in a sentence that was used in all analyses. To make sure our definition of curvature is similar to (Hénaff et al., 2019), which would facilitate relating our findings to the prior work that we are building on, we adopted their metric and simply equated words (tokens) to visual frames . We used tokens for all analyses except for the analysis where we related curvature to n-gram word surprisal, in which case we used words instead of tokens. Given a sequence of words, $w_1, w_2, ..., w_n$, we first extracted the activation weights from each layer of the network, starting from the first contextualized layer $L_0$. Considering layer $L_p$ activation as states $x_1^p, x_2^p, ...x_n^p$, we then computed vectors $v_1^p, v_2^p, ...v_{n-1}^p$ as the difference between to adjacent states $v_k^p = x_{k+1}^p - x_k^p$ (**Figure 1B**). We calculated **curvature** as the angle between these vectors, namely:

$$c_k^p = arccos\left(\frac{v_{k+1}^p . v_k^p}{||v_{k+1}^p||\, ||v_k^p||}\right)$$

We computed *average curvature* across the sentence.

$$C_{s_n}^p = \frac{1}{k}\sum_{i=1}^{k} c_i^p$$

We then computed a *change in curvature* for each sentence between each layer P and the first layer, obtaining a value for each layer:

$$\Delta C_{s_n}^P = C_{s_n}^P - C_{s_n}^1$$

Finally, we computed the average *change in curvature* across all sentences for each layer $P$ as:

$$\Delta C^P = \frac{1}{N}\sum_{j=1}^{N} \Delta C_{s_n}^P$$

### 2.3 Sentence corpus

We used the Universal Dependencies corpus (de Marneffe et al., 2021) to sample a diverse set of sentences.The corpus includes texts on diverse topics that come from books, newspapers, and web-based sources. We filtered sentences to only include 100K most common nouns in English Brants and Franz, 2006, and additionally removed abbreviations and capitalized words. In addition, to ensure that we have sufficient sensitivity to estimate a curvature value per sentence, the sentences were constrained to be between 6 and 19 words. We then used the resulting 8408 sentences to extract model representations, and refer to this corpus as **UDsubset8408**.

### 2.4 Model training

In order to investigate the effects on representational geometry of model training, we trained a GPT2 (12 layers) model with a context window of 1024 using the GPT-NEOX library which is a distributed

training framework that utilizes the DeepSpeed library (Aminabadi et al., 2022; Black et al., 2022). For the training datasets, following Hosseini et al., 2022, we combined BookCorpus and English Wikipedia (Zhu et al., 2015; Liu et al., 2019) in a 1:3 ratio, and created 4 different datasets consisting of 1 million, 10 million, 100 million, and 1 billion tokens. The details of the training are similar to Hosseini et al. (2022). In particular, models with random weight initialization were trained on the next-word prediction objective, with context size of 1024 tokens, and batch size of 128, on 4 NVIDIA RTX A6000 GPUs, with maximum training duration of 1 week. We trained each model until it reached its best validation loss for next-word prediction, with the same validation dataset used across models. (In addition to the critical goal of evaluating the effects of model training on representational geometry, this analysis helps evaluate the robustness of the main results to implementation details.)

### 2.5 Model sequence generation

In order to test whether models favor straight trajectories, we compared the curvature for a set of natural, human-generated sentences and sentences generated by the model from the same initial prompt. To perform this experiment, we selected a subset of the UDsubset8408 corpus (see Sentence Corpus) such that each sentence contained at least 10 tokens, which amounted to 5815 sentences. These sentences (cut off at 10 tokens) constituted the *ground truth* condition. To create the *model-generated* condition, we provided the first 3 tokens to the model, and allowed the model to generate the remaining 7 tokens in a greedy fashion (i.e., by having the model select the token with maximum probability at each step).

### 2.6 Relating sentence surprisal to curvature

To investigate the relationship between curvature and a behaviorally relevant measure of human language processing, we turned to surprisal. Surprisal, a measure of how unexpected a word is in a particular context, has been shown to relate to comprehension difficulty in both behavioral investigations and brain imaging studies (e.g., (Levy, 2008; Smith and Levy, 2013; Willems et al., 2016; Henderson et al., 2016; Lopopolo et al., 2017; Shain et al., 2020; Heilbron et al., 2022)). For each of the sentences in the UDsubset8408 corpus, we computed an average sentence surprisal using the 3-gram measure (Brants and Franz, 2006; Piantadosi et al., 2011).

$$Surprisal(w_n|w_{n-2}, w_{n-1}) = -\log_2 P(w_n|w_{n-2}, w_{n-1})$$

We then computed a Pearson correlation between average sentence surprisal and average sentence curvature for each layer of the model.

## 3 Results

### 3.1 Experiment 1. The curvature of sentence representations decreases across the model layers.

We first tested whether the trajectory of sentences becomes progressively more straight in the internal states of the model across layers. We computed the average sentence curvature for the 8,408 sentences in the UDsubset8408 set across all layers of the GPT2-XL model (see Methods). In **Figure 2A**, we plot for each layer (column), the difference in curvature relative to the curvature in the first layer for each sentence (row). We indeed observed a substantial drop in curvature from the early to the middle layers (as evidenced by darker colors in the middle). Beyond the middle layers, where we see a reduction in curvature for all sentences, the curvature trajectories are somewhat variable across sentences: for some sentences, the curvature remains relatively constant from the middle to the late layers, for others, it increases towards the values in the early layers. The increase in curvature (on average) in the later layers is plausibly due to the fact that the model needs to eventually map its representations back to words in the output, and the word space is inherently nonlinear.

To ensure that curvature reduction is not due to the model architecture alone, we performed the same analysis across the layers of an untrained GPT2-XL model (**Figure 2B**). We did not observe any systematic change in the curvature values across the layers. To better illustrate the difference in the curvature patterns between the trained vs. the untrained model, we selected 300 sentences with maximum curvature drop (the biggest change between the first layer and any of the subsequent layers) for the trained model and 300 sentences with maximum curvature drop for the untrained model. The

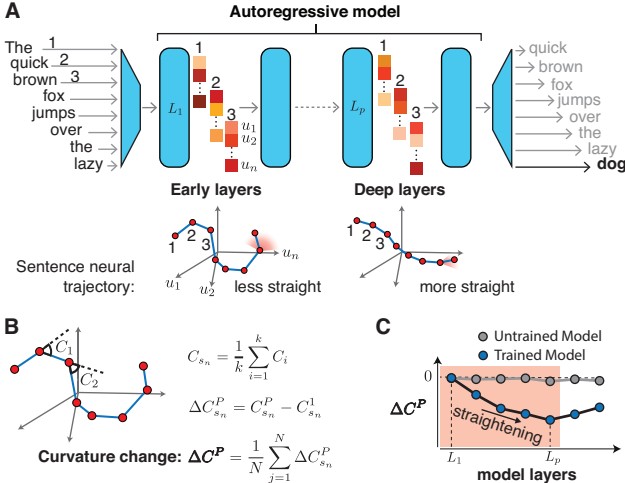

Figure 1: **A.** Top: The structure of autoregressive language models, like GPT2, used in the current study. The representation for each word (token) is extracted from each individual layer of the model. Bottom: The putative sentence trajectory in the state space of the model shown for an earlier vs. a deeper layer. In early layers, the trajectories are less straight, which makes the next state more difficult to predict (depicted by a wide angle in the shaded red region). In deeper layers, the trajectories are more straight,which makes the next state easier to predict (depicted by a narrower angle in the shaded red region). **B.** Sentence curvature computation. The graph shows a putative trajectory for a sample 8-word-long sentence in the multi-dimensional representational space of the model. For each sentence in each layer, we computed sentence curvature as the average of the angles between the vectors that connect adjacent words (C1, C2, and so on). We then computed a change in sentence curvature between each layer and the first layer. Finally, we computed the average change in curvature between each layer and the first layer across a large set of natural sentences (see Methods). **C.** The predictions of the representation straightening hypothesis for trained (blue dots) versus untrained (gray dots) models across layers. The y-axis represents the amount of curvature change between each layer and the first layer (lower values correspond to a greater change in curvature). The hypothesis predicts that for trained, but not untrained, models, the curvature of sentences should decrease from the early to the deeper layers so as to enable efficient next-word prediction (the red-shaded portion of the graph). The curvature may then go back up for the layers that are closest to the output layer because the word space is highly nonlinear.

curvature values show a clear separation between the trained and the untrained model after layer 10 for these sentences (**Figure 2C**), as well as for the full set of 8,408 sentences (**Figure 2D**). Overall then, we found a robust reduction in the curvature of sentence representations, from early to middle layers of GPT2-XL, and only for the trained version of the model.

### 3.2 Experiment 2. The curvature of sentence representations decreases to a greater extent in larger models and with more training.

In the literature on foundation models, model performance on the next-word prediction task scales proportionally to the model size and the training dataset size (Kaplan et al., 2020). We tested whether curvature reduction may provide a mechanistic-level explanation for these relationships in terms of internal model states.

To test the effect on curvature of **model size**, we computed the average sentence curvature for the same set of 8,408 sentences used in Experiment 1 across all layers of a class of GPT2 models that vary in the number of parameters (82, 117, 345, 774, and 1,558 million parameters). Replicating the basic finding for GPT2-XL from Experiment 1, we observed a drop in curvature from the early to the deeper layers in the four smaller models (GPT2-large, GPT2-medium, GPT2, and distillGPT2; **Figure 3A**). Critically, larger models exhibited larger decreases in curvature. The average change in curvature shows a logarithmic relationship with model size (**Figure 3B**).

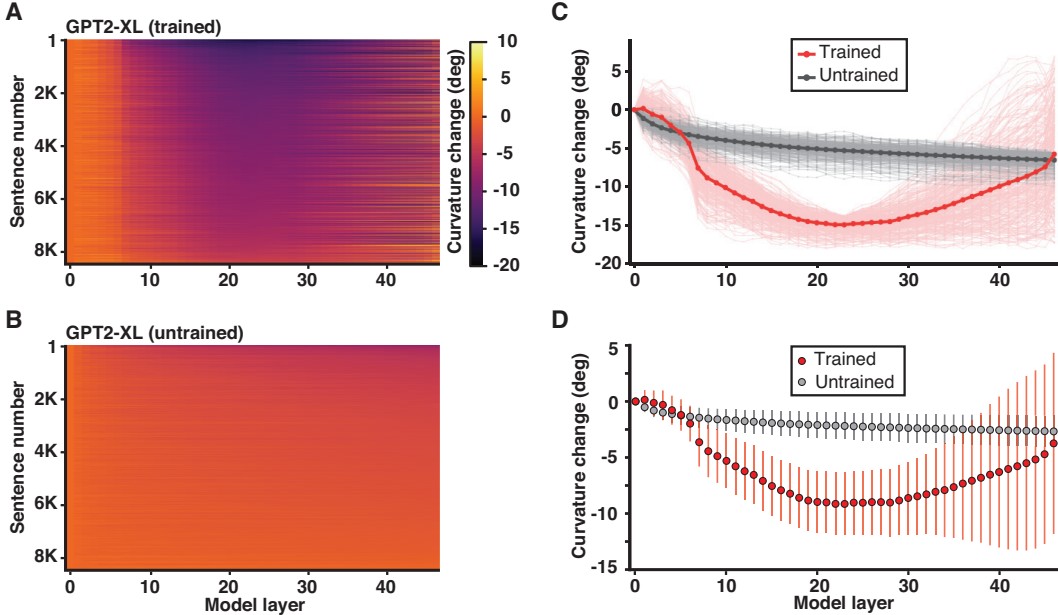

Figure 2: **A-B.** Curvature changes (relative to the first layer) across the layers of the network (GPT-2-XL) (columns) for sentences in UDsubset8408 set (rows). For the trained model (A), a consistent drop in the curvature is observed from the early to middle layers of the network. No such drop is seen for an untrained GPT-2-XL model (B). **C.** Curvature changes across the layers of the trained and untrained network (red and gray dots, respectively) for a set of 300 sentences selected separately for the trained and untrained models as having a maximum curvature drop. Individual lines correspond to sentences. A clear separation between the trained and untrained model is observed after layer 10. **D.** Average curvature changes across the layers of the trained and untrained network (red and gray dots, respectively) for all sentences in UDsubset8408 set. Error bars show standard deviation over sentences in each layer.

To test the effect on curvature of **training corpus size**, independent of model size, we trained four GPT2 models (12 layers) using datasets with a controlled number of words, similar to Hosseini et al., 2022. The datasets were scaled logarithmically in size (1 million, 10 million, 100 million, and 1 billion words). After training, we computed the average sentence curvature for our set of 8,408 sentences across all layers of each model. As can be seen in **Figures 3C-D**, the model trained on 1 million words exhibits a minimal change in curvature (close to an untrained model). The models trained on 10 million and 100 million words exhibit progressively larger decreases in curvature. Moreover, the layer with the largest curvature reduction shifts from the earlier to the deeper layers as a function of training corpus size. Interestingly, the model trained on 1 billion words does not show further reduction in curvature, reaching a similar level of curvature reduction as the model trained on 100 million words.

Thus, both model size and training corpus size affect the curvature of sentence representations, such that larger models and models trained on more data achieve straighter representations in the deep model layers. We hypothesize that, mechanistically, the greater degree of representation straightening is what leads to better next-word prediction performance in larger models and models trained on more data.

### 3.3 Experiment 3. The model favors straight trajectories during language generation.

If models rely on representation straightening to make predictions about upcoming words, then the trajectories of sentences that are generated by the model should be straighter than natural, human-produced sentences, given that next-word prediction is not the only objective that guides human language production (**Figure 4A**). This constitutes perhaps the most direct test of the representation straightening hypothesis. To test whether models favor straight sentence trajectories, we designed a

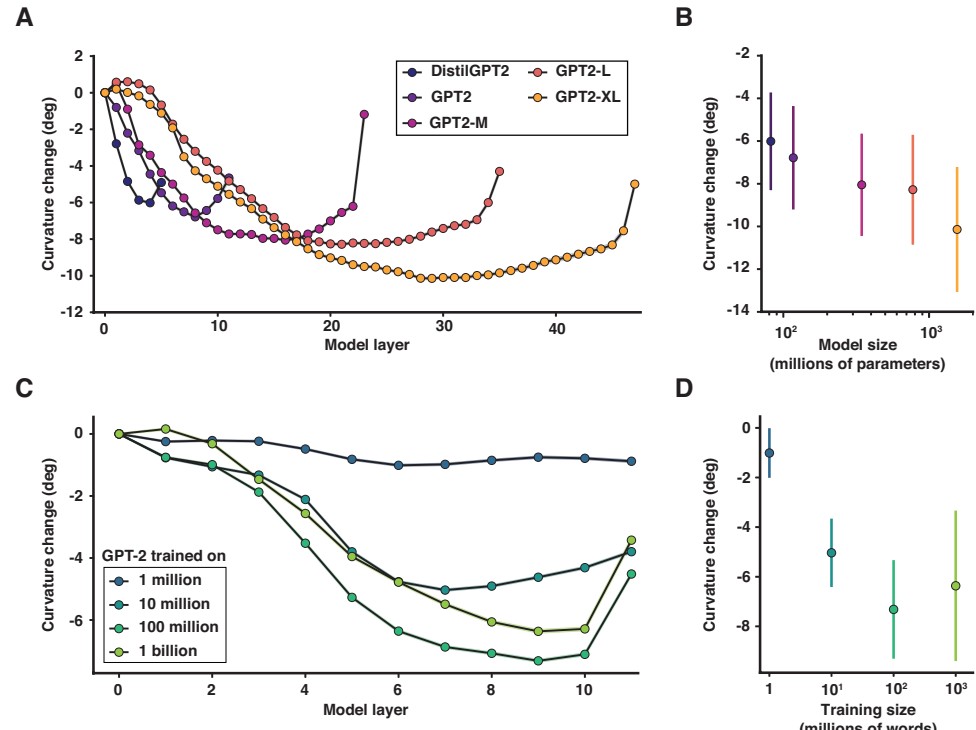

Figure 3: **A.** Curvature changes (relative to the curvature in the first layer) across the layers of the network for five GPT2-class models of different sizes (each model is a line; darker lines=smaller models; each dot is a layer) for the 8,408 sentences in the UDsubset8408. For all models, a consistent drop in curvature is observed from the early to deeper layers of the network. However, larger models, with better next-word prediction performance, exhibit greater curvature reduction. **B.** The relationship between model size and curvature change for the layer with the largest average curvature reduction. **C.** Curvature changes across the layers of the network for four versions of GPT2 (gpt-neox implementation; Black et al., 2022) trained on different-size corpora (each model is a line; darker lines=models trained on smaller corpora; each dot is a layer). For all models except for the one trained on the smallest corpus (1M tokens), a consistent drop in curvature is observed from the early to deeper layers of the network. Models trained on more data exhibit greater curvature reduction, but this effect appears to plateau with datasets larger than 100M tokens. (Note that here we examine curvature changes across layers (relative to the first layer); absolute curvature continues to decrease for larger training datasets (not shown). **D.** The relationship between training corpus size and curvature change for the layer with the largest average curvature reduction.

controlled experiment. We first selected a subset of sentences from our set of 8,408 sentences that consist of at least 10 tokens (n=5,815 sentences). These corpus-extracted sentences constitute our *ground-truth* condition. We then created alternate versions of these sentences by providing the model (GPT2-XL) with the first 3 tokens of each sentence and allowing the model to generate a sequence of 7 tokens as a continuation (*model-generated* condition; for example sentence pairs, see **Figure 4B**). We then compared the curvature for the ground-truth sentences (cutting them off at 10 tokens) vs. the model-generated sentences.

The pattern of curvature change is similar between the two conditions in the early layers. Critically, starting around layer 7, the model-generated sentences exhibit a larger drop in curvature, and this between-condition difference increases over the subsequent layers, peaking around layer 20 (**Figure 4B**). The sharper decrease in curvature for the model-generated sentences is predicted by the representational straightening hypothesis. These results also generalize to smaller models and to prompts of different lengths (not shown here).

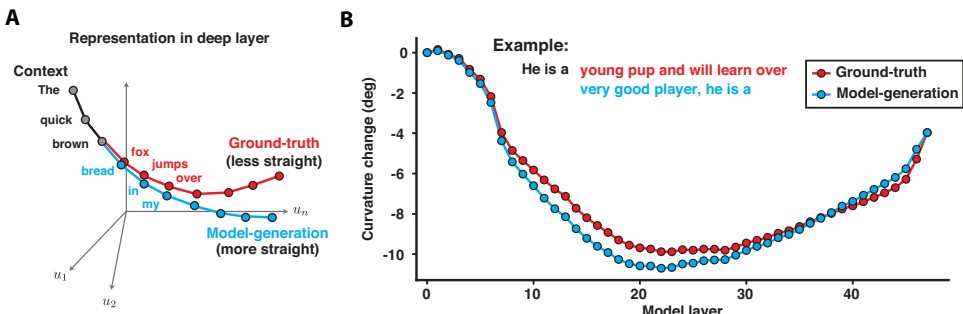

Figure 4: **A.** The predictions of the representation straightening hypothesis for the ground-truth (natural, human-produced) sentences (red line) vs. model-generated sentences (blue line). The hypothesis predicts that the trajectories of the model-generated sentences should be straighter given that linear extrapolation in the internal state space is hypothesized to serve as the critical prediction mechanism. In other words, if the model is internally producing a low-curvature trajectory, then self-generated sequences should have lower curvature than sequences generated by humans (given that next-word prediction is not the only objective that guides human language production). **B.** Curvature changes across the layers of the network for 5,815 pairs of 10-token sentences: the ground-truth sentences (red line; each dot is a layer) come from the UDsubset8408 set and the model-generated sentences (blue) are generated from the prompt consisting of the first three tokens of the ground-truth sentences (the model generates the subsequent 7 tokens using a greedy approach; Methods). Model-generated sentences show a greater drop in curvature reduction relative to the ground-truth sentences. An example of ground-truth vs. model generated sentence is shown on top of the panel.

### 3.4 Experiment 4. The curvature of sentence representations is correlated with sentence surprisal.

In Experiments 1-3, we have focused on language models and established that models reduce the curvature of sentence trajectories in their internal state spaces. Using a set of 8408 sentences, we found consistent curvature reduction in the deep model layers across this corpus. However, we also observed some variability among the individual sentences in their curvature patterns across the layers (**Figure 1C**). In Experiment 4, we attempted to connect this variability in the geometry of sentence representations in the model to some linguistic features of the sentences that we know affect human language processing. In particular, we focused on surprisal—a measure of how expected a word is given context. Surprisal has been shown to affect behavioral (Levy, 2008; Smith and Levy, 2013) and neural (Willems et al., 2016; Shain et al., 2020) responses to language. So we asked whether average sentence surprisal (we used 3-gram surprisal, averaging across the words to derive a single measure for each sentence; see Methods) relates to the curvature of the sentence's trajectory in the model space. If higher curvature in the model space corresponds to greater difficulty in predicting the next word, we should observe a positive relationship between curvature and surprisal: sentences that are overall less predictable (more surprising) should have higher curvature. Moreover, this correlation should only be observed in deeper layers (**Figure 5A**).

The results are shown in **Figures 5B-C**. As expected, for the untrained model, we found no relationship between surprisal and curvature (0 correlation across layers, except for layer 0). However, for the trained model, the correlation starts to increase from the early layers toward the deeper layers, peaking around layer 20 (**Figure 5C**). This result suggests that the degree to which the model straightens a sentence trajectory internally is associated with how surprising the sentence is behaviorally. These results also generalize to smaller models (not shown) and to other surprisal metrics (other n-gram metrics and PCFG-parser-based surprisal; see Supplementary Information).

## 4 Discussion

In this work, building on Hénaff and colleagues' proposal for primate vision (Hénaff et al., 2019; Hénaff, 2018; Hénaff et al., 2021), we established **neural trajectory straightening** as a representational hypothesis about how neural network language models perform next-word prediction. Models consistently reduced sentence curvature from early to middle layers, and this effect was only observed

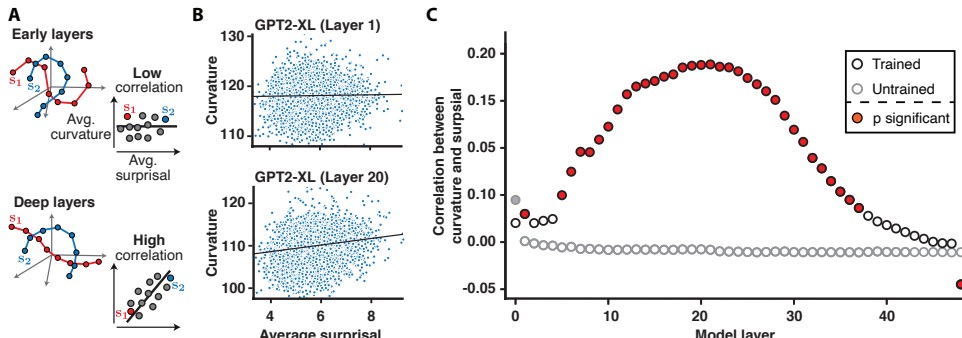

Figure 5: **A.** A hypothesized relationship between curvature and surprisal in the early layers (top) and the middle layers (bottom) of the network. In early layers the representations are close to input, so the curvature is not correlated with sentence surprisal. In deep layers however the curvature is predictive model states, so there is a consistent correlation between the sentence curvature and sentence surprisal. **B.** The relationship between average sentence surprisal (3-gram surprisal; see Methods) and average sentence curvature for sentences in the UDsubset8408 set in layer 1 (top) vs. layer 20 (bottom). The lines show a linear regression fit to the data (for illustrative purposes). **C.** Pearson correlation between average sentence surprisal (3-gram) and average sentence curvature across the layers of the network for an untrained model (gray-contour dots) and a trained model (black-contour dots). The data points for the layers where the correlation reached significance are color-filled (grey for untrained; red for trained). For the untrained model, no relationship is observed in any of the layers, except for layer 0. For the trained model, there is a consistent increase in correlation between curvature and surprisal from the early to middle layers.

for trained models (**Figure 2**). Model size and training dataset size affected the model's ability to reduce curvature (**Figure 3**), and model-generated sentences exhibited lower curvature compared to natural human-generated sentences (**Figure 4**). Finally, average sentence curvature correlated with average sentence surprisal in the middle layers of the model **Figure 5**).

Our results sit squarely within the efficient coding framework and establish temporal prediction as a form of efficiency that is born out of training autoregressive transformer models on next-word prediction. These findings may also explain why representations from the middle layers of transformer language models align best with human neural responses during language processing (e.g., Schrimpf et al., 2021; Goldstein et al., 2022; Caucheteux and King, 2022; Caucheteux et al., 2023; Toneva and Wehbe, 2019; Jain and Huth, 2018).

Using representational geometry to understand the relationship between the internal workings of artificial and biological systems and their behavior has gained momentum in recent years( Wang et al., 2018; Remington et al., 2018; Mante et al., 2013; Chung et al., 2018 for a review, see Chung and Abbott, 2021). For example, in a deep neural network of vision, Cohen et al., 2020 showed how the geometry of object manifolds changes across model layers and how it relates to model performance in object categorization. In the domain of lang, Hewitt and Manning, 2019 found that the representations in the middle layers of BERT best capture the hierarchical structure of sentences In another line of work, Mamou et al., 2020 used manifold analysis to uncover how different features of words (e.g., part of speech) and sentences become separable in the deep layers of language models like BERT and GPT. More recently, Valeriani et al., 2023 investigated how geometric properties, such as intrinsic dimensionality, change across the layers of the transformer models, and found that dimensionality increases across layers before sharply decreasing in deeper layers of the bidirectional transformer models. These prior studies provide insights into the geometric properties of vision and language models, but, to our knowledge, no prior study has evaluated a representational-level hypothesis that connects language model behavior (next-word prediction) to neural trajectories of individual sentences.

In the representation straightening hypothesis, the problem of predicting the next token/word is reformulated as predicting the next state in the model's internal representation. Upon predicting the next state, the model can connect this new state to the features of the input/output (i.e., to words). Explicitly defining model's objective to build a predictive representation over its internal

representation, and not output, and would be an interesting future direction (Olshausen and Field, 1996). Another direction would be to evaluate representation straightening in human behavioral or neural responses to language. Finally, if representation straightening is a general mechanism for temporal prediction, it should be evident in other systems, biological and artificial, that have a core prediction objective; it would be important to evaluate the generality of this mechanism.

It is important to note that in the representation straightening hypothesis, the problem of predicting the next token/word is reformulated as predicting the next state in the model's internal representation. Upon predicting the next state, the model can then connect this new state to the features of the input/output (i.e., to words). Explicitly defining such transformation stages, would allow the model to build a predictive representation without any apparent behavior and would be an interesting future direction. Another exciting direction would be to evaluate representation straightening in human neural language data direction. Finally, if representation straightening is a general mechanism for temporal prediction, it should be evident in other systems, biological and artificial, that have a core prediction objective; it would be important to evaluate the generality of this mechanism.

## 5   Broader Impact and Limitations

Our work puts forward and provides support for a general hypothesis at the representational level about a mechanism that allows large language models to achieve good performance on next word prediction and potentially downstream tasks. This work adds to the growing body of research on the interpretability of AI models. A better, more mechanistic understanding of these models, and potentially other models with prediction objectives, can both i) suggest ways to improve model efficiency and robustness, and ii) provide insights into the relationship between neural networks and the human language system.

We acknowledge that our work could be improved in several respects. The results as they stand are compatible with at least two possibilities: (i) that predicting future inputs intrinsically gives rise to implicit next-state prediction, thus directly favoring linear state dynamics, or (ii) that predicting future inputs in domains like language benefits from slowly-changing contextual information, thus indirectly favoring slower (and more approximately linear) state dynamics. These possibilities can be distinguished in the future by training models on artificially created datasets that vary in the length of context that affects the predictability of an incoming element. If the effects we report here obtain across these different training datasets, that would support the first possibility; if instead the effects only hold for models trained on data where relatively long predictive contexts, that would support the second possibility.

Furthermore, we have not evaluated the effects on sentence curvature of other training objectives or fine-tuning for downstream tasks. Doing so can help understand the *selectivity* of the observed effects (i.e., do sentence representations get straightened in the middle layers only under the pressure of the next-word prediction objective?) and their *robustness* to adding other objectives to a pre-trained model. We have also not *causally* tested the straightening hypothesis, which would require ablating the model in such a way that only curvature is affected, and testing how next-word prediction behavior changes.

It is also important to note that we are not claiming that representation straightening is the only mechanism that models rely on to gain linguistic competence. However, to the extent that prediction is a core part of language learning and processing (in artificial as well as biological systems), we are showing that targeted inspection of geometric properties of sentence representation gives rise to a hypothesis about how prediction may be implemented in language models.

## 6   Acknowledgements

The authors would like to thank Eero Simoncelli, Olivier Hénaff, Yoon Bao, and Cory Shain for helpful insights and discussions, Anya Ivanova, Chengxu Zhuang for feedback on the manuscript, as well as members of EvLab at MIT. EF was supported by NIH grants R01-DC016950 and U01-NS121471, and by funds from the McGovern Institute for Brain Research, the Brain and Cognitive Science Department, the Simons Center for the Social Brain, and MIT Quest for Intelligence.

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
