

**Supplementary figure 1.** Ablation study, each row represents the ablated layer and each column the module that is ablated from that layer, for example the first panel shows ablation of attention-key in layer 5. Different layers in GPT2-XL model were ablated and the consequence of ablation on curvature measured for 2000 sentences in UD corpus. Red bar shows the layer where ablation was applied. Ablation is most effective when performed on attention module and in early layers of the model

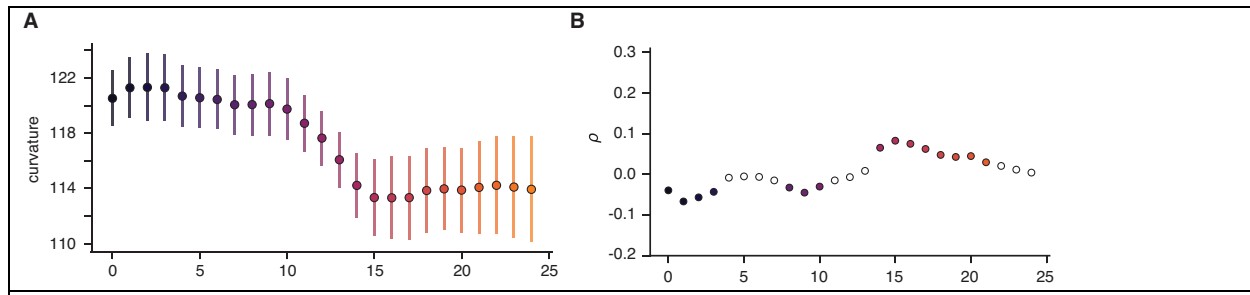

**Supplementary figure 2.** A. curvature values for BERT-large-uncased model across all sentences. B. correlation between 3-gram surprisal and curvature in BERT-large-uncased

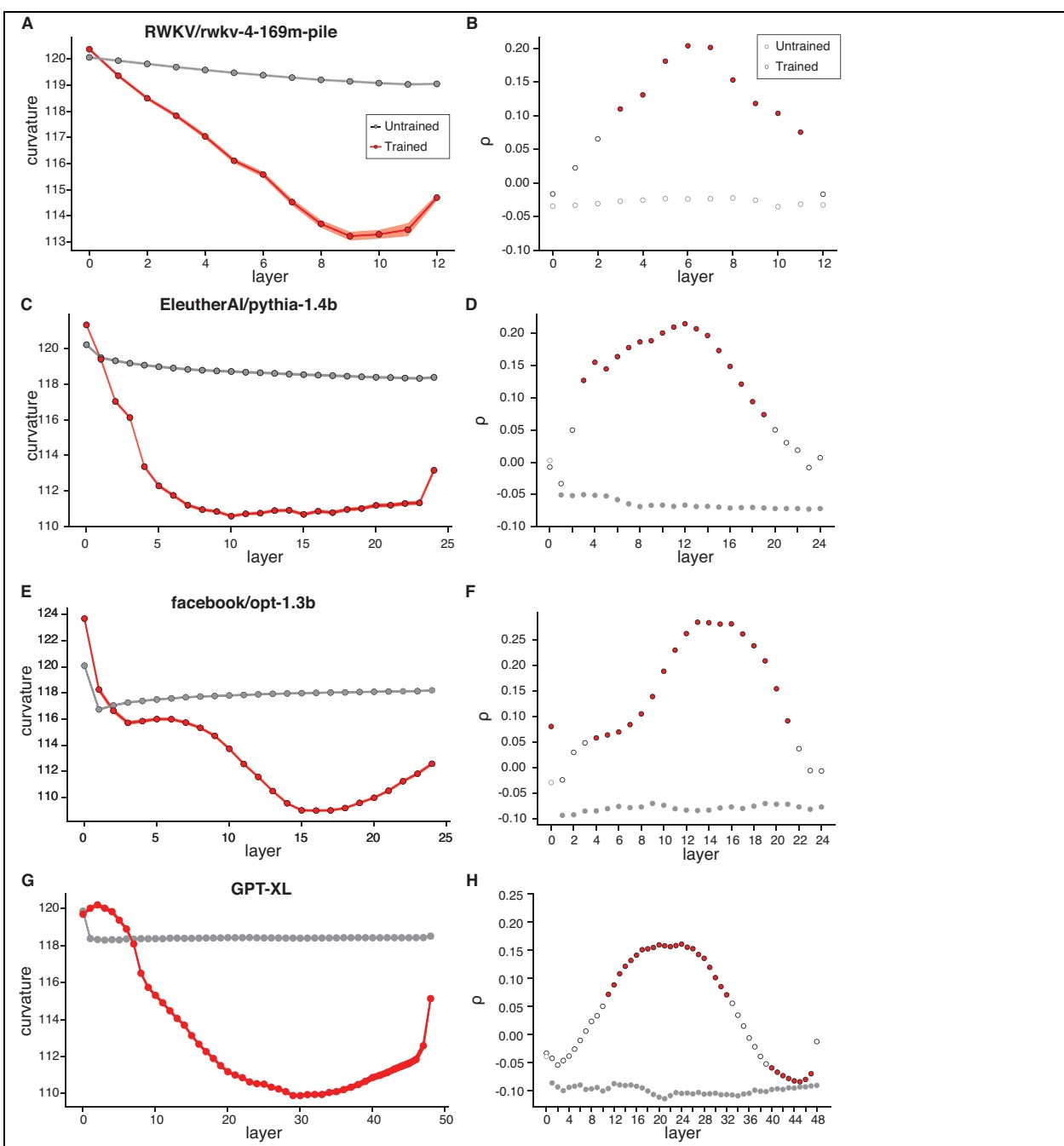

**Supplementary figure 3. A.** curvature values for sampled 2000 sentence in RWKV model ( RNN) for both trained an untrained version. **B** correlation between model generated surprisal and curvature in RWKV model. **(C,D),** same as A and B for EleutherAI/pythia-1.4b model ( transformer with rotary positional encoding **(E,F)** same as A,B for Facebook/opt-1.3B model. **(G,H)** same as A,B for GPT2-XL model.

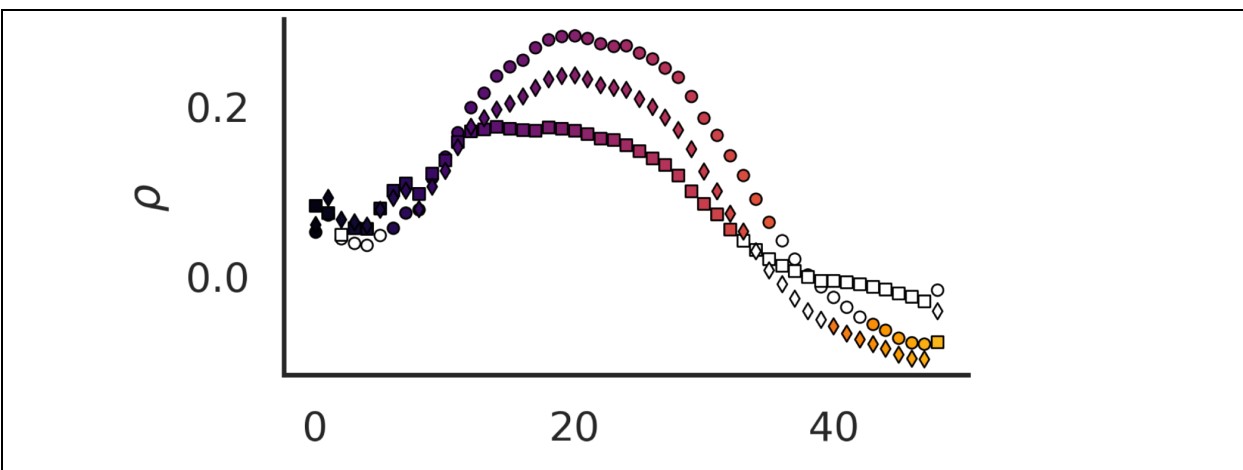

**Supplementary figure 4.** Relationship between curvature in GPT2-XL and surprisal for different types of surprisal estimate. Circles: 3-gram, squares: lexical. Diamonds: syntactic surprisal

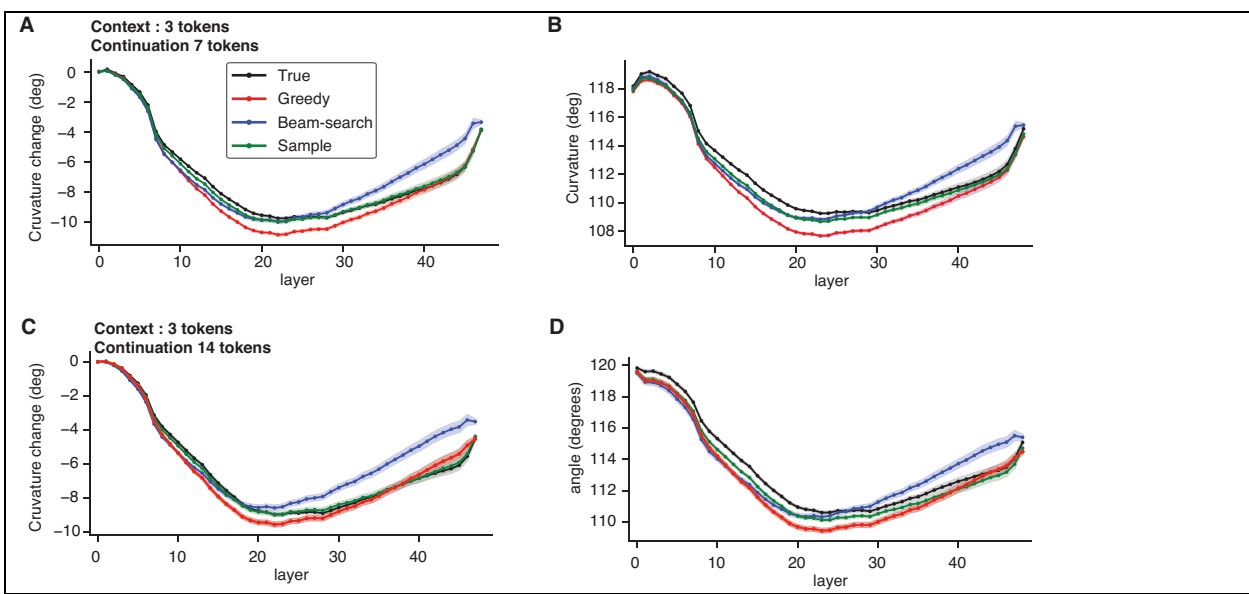

**Supplementary figure 5:** Effect of different decoding strategies in GPT2-XL sequence generation and its comparison to ground-truth(true) same as figure 4b in the main manuscript. **A** curvature change values. **B** curvature values. **C,D** same as A and B but for context of 3 and continuation of 14 tokens.

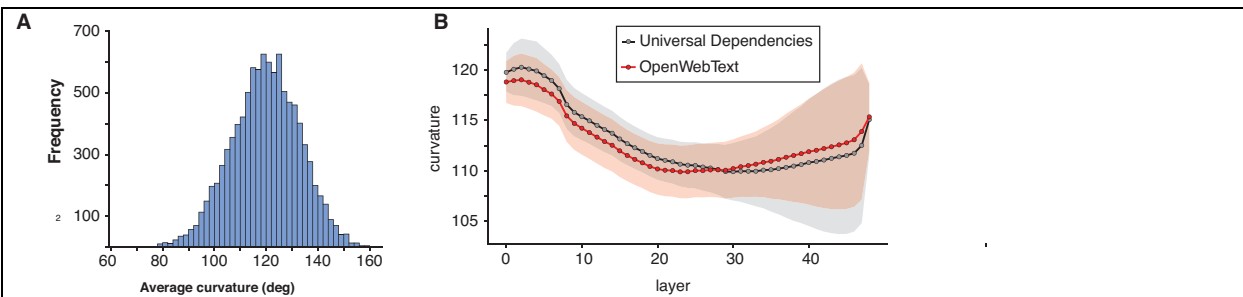

**Supplementary figure 6. A** curvature for a set of random trajectories created with the same dimensionality as GPT2-XL. The distribution of points is close to 120 degrees. **B** curvature values for GPT2-XL for 2 different datasets. Shaded region show standard deviation over 8408 sentence in Universal dependencies (UD), and 8408 sequences in openwebtext sampled randomly and then cut to match the length of sentences in UD corpus, ( some sequences might night be full sentences)

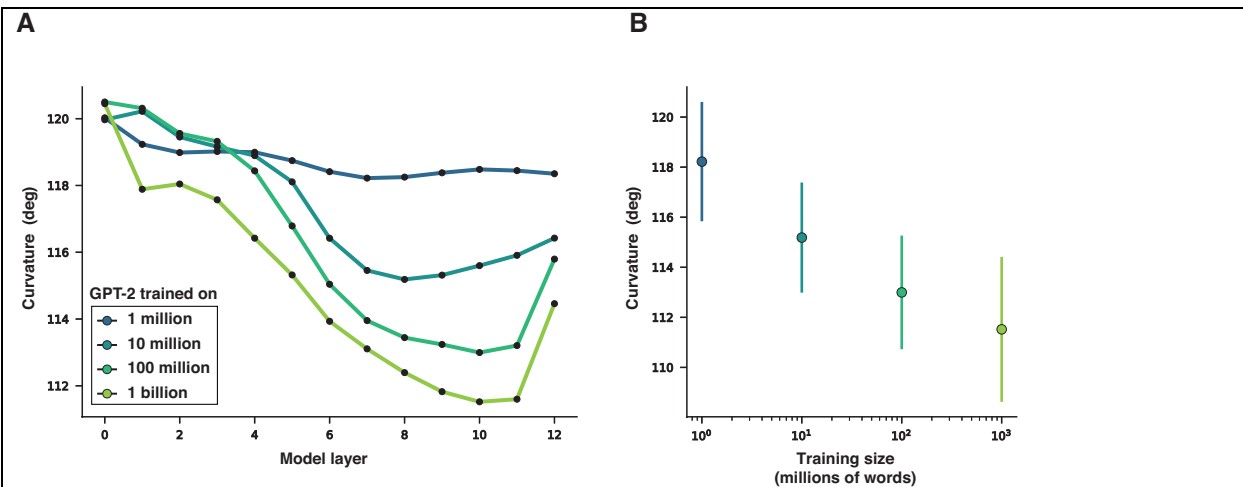

**Supplementary figure 7**: effect of training on curvature for GPT2 model with 12 layers. Models were train on 1 million,10 million, 100 million and 1 billion words. **A.** relationship between curvature and model layer for each training amount. **B.** lowest curvature achieved in each model (corresponds to different layers for different training amount.