# OpenReview forum: "Large language models implicitly learn to straighten neural sentence trajectories to construct a predictive representation of natural language."
_NeurIPS.cc/2023/Conference — NeurIPS 2023 poster_

### Official Review · Reviewer_TdxE · 2023-07-06

**Soundness:** 3 good
**Presentation:** 3 good
**Contribution:** 3 good
**Rating:** 6
**Confidence:** 4

**Summary:**

In transformers architecture a word sequence is first represented by a sequence of word embeddings. This sequence of vector is then iteratively transformed by the successive transformer layers of the model. This paper proposes to characterize the trajectories taken by word embedding sequences. Features are introduced to better understand what happens during these transformations: the curvature along one dimension is estimated by looking the angle of the "curve" (the arcos) and trying to make correlation with "surprisal".


**Strengths:**

It is important to understand what is going on inside transformers and what kind of transformation are learnt by the model. Looking at the trajectories is clearly a good idea. This notion of curvature tries to characterize the surprisal observed in the data based on a intuitive assumption. Some experimental results show that the curvature can indeed capture something of the iterative transformations.



**Weaknesses:**

While the starting idea is nice, the submission needs to be improved. Many important points remain unclear. Here is a list in reading order (more or less).

- Concerning the dataset: UD is a multilingual dataset, what is the language ? Why selecting only very short sentences ?
- For the models used, it is really messy. Are the models retrained from scratch ? finetuned ? All along the paper, we never know.
- On the same topic, what does "untrained model"  mean ? Is it randomly initialized ? Only pre-trained ? Why it is a good basis for comparison ?
- This a bit similar for evaluation data: you could define the different datasets once and make clear reference afterwards.
- The decoding strategy maybe importance for some measurement. Maybe the greedy choice is not the best.
- The definition of surprisal could be given and related to the perplexity/NLL which can be the optimization criterion.
- At the end the statistical observation are not so impressive, or I did not clearly understand. The claim are not so clear at the end. It is difficult to really conclude that the prediction aims at  linearizing the input through a series of transformations.

- You could use latex reference with section numbers (see eg line 130).
- The colors used for some figures (3 and 5) are really difficult to distinguish even on a color printed version, or with the pdf.
- The figure captions are very long !


As a conclusion, I really liked the starting idea of the paper and I think that it deserves further improvement before submission.


**Questions:**

The notion of straithening is a nice idea and a important contribution
of the paper. Maybe it deserves a discussion.  Why it is estimated
like that ? Is there other features that can characterize the
curvature ? Maybe they are less tractable, but they are important. It
could be nice to relate this notion to the manifold defined by the
trajectories.

Why the change in curvature is only measured relatively with the first layer ?
Why only looking at the cosinus ?  The cosinus cannot distinguish the sign of the angle.

---

> ### Author Rebuttal · Authors · 2023-08-10
>
> Weaknesses:
> While the starting idea is nice, the submission needs to be improved. Many important points remain unclear. Here is a list in reading order (more or less).
>
> Concerning the dataset: UD is a multilingual dataset, what is the language ? Why selecting only very short sentences ?
> We focused on only English sentences in UD corpus. We limited the number of words in the sentence so that at we don’t get averaging effect since we are looking at an average metric. we used universal dependencies because we wanted to be able to quantify the sentences and phrases along multiple lexical, semantic and syntactic feature. UD corpus provides precomputed values for syntactic features. We further restricted our sentences to include 100K most common words in English (using google ngram dataset). Our choice of sentence length was to ensure we have enough sensitivity in measuring average curvature. Sentences that are very long would result in averaging out the differences that we are interested in.
>
> For the models used, it is really messy. Are the models retrained from scratch ? finetuned ? All along the paper, we never know.
> We are sorry for this overlooking this. Yes, the models were trained on next word prediction objective from scratch. We will address this in the updated version of the paper
>
> On the same topic, what does "untrained model" mean ? Is it randomly initialized ? Only pre-trained ? Why it is a good basis for comparison ?
> The untrained model is indeed randomly initialized (same procedure as (Radford et al.,2020.)) we believe it is a good basis for comparison because the representation is not yet shaped by the training objective, and only shaped by the architecture and operations such as layer norm. the lack of evidence for curvature in untrained model further suggests that it is a consequence of predictive objective that the model is trained on.
>
> This a bit similar for evaluation data: you could define the different datasets once and make clear reference afterwards.?
> We will update the manuscript accordingly
> The decoding strategy maybe importance for some measurement. Maybe the greedy choice is not the best.
> For completeness we have included in supplementary figure 5 additional decoding strategies. However for the purpose of interrogating the internal representation of the model, we think the greedy is the most direct one.
>
> The definition of surprisal could be given and related to the perplexity/NLL which can be the optimization criterion.
> We included a supplementary figure 3G,H to address reviewer’s concern. We observed similar relationship between surprisal and curvature for model generated surprisal.
>
> At the end the statistical observation are not so impressive, or I did not clearly understand. The claim are not so clear at the end. It is difficult to really conclude that the prediction aims at linearizing the input through a series of transformations.
> We would appreciate if the reviewer could point us to specific observations so that we could address their concern.
>
> You could use latex reference with section numbers (see eg line 130).
>
> The colors used for some figures (3 and 5) are really difficult to distinguish even on a color printed version, or with the pdf.
> We appreciate this comment and can address specific figures that the reviewer think would benefit from restructuring.
>
> The figure captions are very long !
> We will update the figure captions in the updated manuscript to fix this.
> As a conclusion, I really liked the starting idea of the paper and I think that it deserves further improvement before submission.
>
> We thank the reviewer for the positive outlook, and will integrate all their input to improve the work before submission
>
> Questions:
> The notion of straithening is a nice idea and a important contribution of the paper. Maybe it deserves a discussion. Why it is estimated like that ? Is there other features that can characterize the curvature ? Maybe they are less tractable, but they are important. It could be nice to relate this notion to the manifold defined by the trajectories.
> We think the current estimation is the simplest way of parametrizing the neural sentence trajectory. We agree that connecting trajectories to the manifolds that govern their evolution in the internal representation of the model would be an exciting next direction
>
> Why the change in curvature is only measured relatively with the first layer ?
> Why only looking at the cosinus ? The cosinus cannot distinguish the sign of the angle.
> We measured the curvature change from the first layer to be able to compare models. We have included original curvature values in the supplementary figure.
> We always observed positive curvatures in model representations, as can be seen in the supplementary figures 1, 2, 3,and 6

---

### Official Review · Reviewer_WYqK · 2023-07-07

**Soundness:** 4 excellent
**Presentation:** 3 good
**Contribution:** 3 good
**Rating:** 7
**Confidence:** 3

**Summary:**

This paper hypothesizes that the deep, casually-masked transformer models learn to predict by linearizing representational trajectories. This hypothesis is rooted in observations from the neuroscience literature. The hypothesis is tested through experiments that probe:

   (1) the degree to which representation curvature decreases with network depth,
   (2) the relationship between curvature and model performance,
   (3) the curvature of representations of model-generated text,
   (4) the relationship between text surprisal (entropy?) and curvature.

Curvature is defined in the sense of pairwise cosine-similarity between adjacent representations, averaged across sequences of text.

**Strengths:**

The straightening hypothesis is interesting, and the experiments convince me that transformers do exhibit straightening behavior. The experiments appear to be generally well executed (but see my questions below). Experiments #1 and #2 in particular clearly establish the pattern of increased straightening as a function of model depth, model size, and optimization steps.

It seems possible that this observation of straightening could be important and exciting to the neuroscience community, but I do not have the right background to make that judgement.

**Weaknesses:**

It is not clear to me why the straightening hypothesis is important. Accepting that LLM's do indeed straighten trajectories, what should I do with this knowledge? The conclusion gestures at the possibilities with respect to interpretability of models and revealing "when and how they could fail and suggest ways to make models more efficient and robust." But the connection between straightening and these broader goals (which are undoubtedly of relevance to a broader NeurIPS community) are are not clear to me.

I am very open to an argument that more clearly makes the case straightening is important: either from the perspective of its importance to the neuroscience community, or for its potential significance to the broader machine learning community (as hinted at in the conclusion).

**Questions:**

Is there a connection between straightening and Koopman operator theory? For Koopman, non-linear dynamics are explicitly linearized in a latent Hilbert space. Could we view the behavior of trained autoregressive transformers as approximations to a Koopman operator?

Do you believe this result is specific to the transformer architecture? Would we also expect to observe this straightening in state-space models, or causally masked convnets?

Do we know whether text in the Universal Dependencies dataset might have been part of the GPT-2 training data? How might this affect the results if this data was indeed included during training?

Why are the generated sequences for Experiment #2 so short (7 tokens, with a 3 token prompt)?

Why measure surprisal in Experiment #4 using trigrams? Would an information measure like entropy be more natural?

**Limitations:**

As the authors acknowledge, the stronger hypothesis--that straightening is a consequence of the predictive loss--is not supported by the experiments in this paper. This would require experiments involving, e.g., MLM or classification tasks to observe whether straightening also appears in these settings.

---

> ### Author Rebuttal · Authors · 2023-08-10
>
> Strengths:
>
>
> We agree with the reviewer on this point, and to our knowledge, no prior studies in neuroscience have investigated geometrical properties of language networks related to straightening and connecting it behavior of these models.
>
> Weaknesses:
>
> We thank you the reviewer for bringing this issue. We think the building a hypothesis at the level of representation, and not behavior, is the first step in understanding model behavior. Diagnosing failure modes of the model at the level of behavioral is useful but not enough to reveal what change in the internal representation could avoid them. Our work develops a hypothesis instead at the level of representations. In figure 4 for example we showed that model generated sentences diverge from original sentences, and this is evident in the difference in the curvature between the two conditions. One could foresee that using for example a new decoding strategy could make the two curvature more similar and thus the model generate sequences to ground truth text. We demonstrated this in supplementary figure 5, as some decoding strategies are closer to ground-truth than greedy sampling.
> Moreover, the straightening can be used to design more efficient models. For example Olshausen and Field (Olshausen and Field 1996) showed that a sparsity prior over internal representation can push the neural network to develop receptive fields that are similar to that of  biological visual system. We aim to use the straightening as an inductive bias over the model representation and train new models suitable for next word prediction tasks.
>
> We will certainly elaborate on these points in the updated version of the work
>
> Questions:
> Is there a connection between straightening and Koopman operator theory? For Koopman, non-linear dynamics are explicitly linearized in a latent Hilbert space. Could we view the behavior of trained autoregressive transformers as approximations to a Koopman operator?
> This is a very interesting proposal and we thank the reviewer to bring this to our attention. This is beyond the scope of the work, and we can hint towards it in the discussion part of the work
>
> Do you believe this result is specific to the transformer architecture? Would we also expect to observe this straightening in state-space models, or causally masked convnets?
> In supplementary figure 3, we show suggestive evidence that the state-space models exhibit similar behavior when trained on next word prediction objective., and in supplementary figure 2   we showed that bidirectional models behave differently from unidirectional models. Unfortunately, we are not aware of models that are causally masked convnet in text model. We would appreciate if the reviewer could point us to one so we can test their curvatures.
>
> Do we know whether text in the Universal Dependencies dataset might have been part of the GPT-2 training data? How might this affect the results if this data was indeed included during training?
> (search if UD was a part of openwebtext).
> Given that the training dataset for GPT2 is not publicly available, we instead compare model behavior on openwebtext and UD corpus supplementary figure 6, we do observe similar curvature reduction between both datasets.
>
> Why are the generated sequences for Experiment #2 so short (7 tokens, with a 3 token prompt)?
> We chose 3 token as a prompt for 2 reason,
> 1.	We wanted to limit the amount of veridical information the model is exposed to before generating the sequence of tokens
> 2.	The definition of straightening requires at least 3 initial points, so we wanted to give the model the minimal amount of information, in order to have the model produce its prior over trajectories
>
> Why measure surprisal in Experiment #4 using trigrams? Would an information measure like entropy be more natural?
> We included in supplementary figure3G,H correlation between model generated surprisal and curvature, and observed a similar relationship as the main manuscript. We hope this addresses this question
>
> Limitations:
> As the authors acknowledge, the stronger hypothesis--that straightening is a consequence of the predictive loss--is not supported by the experiments in this paper. This would require experiments involving, e.g., MLM or classification tasks to observe whether straightening also appears in these settings.
>
> We agree and in future work we aim to address this.

---

> > ### Comment · Reviewer_WYqK · 2023-08-14
> >
> > Thanks for answering my questions!
> >
> > > Unfortunately, we are not aware of models that are causally masked convnet in text model. We would appreciate if the reviewer could point us to one so we can test their curvatures.
> >
> > Hyena might be a good convolutional model to investigate:
> >
> > https://github.com/HazyResearch/safari
> >
> > Given the results for RWKV, I find it highly likely that the straightening phenomenon is architecture-independent; no need to rush to run additional experiments on my account (although they might be a nice addition to round out the results for the camera ready).
> >
> > The straightening phenomenon is a well-documented by this paper, and apparently it is a general observation about the representations learned by autoregressive language models. It remains unclear to me whether this observation is important, but only time can answer that question. I believe that the observation of straightening is of broad interest to the Neurips community, and I am satisfied by the authors responses to my questions. In light of this, I have raised my score.

---

### Official Review · Reviewer_Sn24 · 2023-07-10

**Soundness:** 3 good
**Presentation:** 3 good
**Contribution:** 2 fair
**Rating:** 5
**Confidence:** 4

**Summary:**

This work investigates whether large language models learn to "straighten" the word-by-word representation of sentence as it passes through the model layers. The word-by-word curvature of the sequence embeddings is defined as the angle between two consecutive word embeddings (i.e. arccos of the cosine similarity of consecutive word embeddings from a particular layer). The idea is that a "straighter" trajectory would enable generalization via extrapolation. This work tests a number of models from the GPT-2 family of various sizes and shows that the word-by-word sequence curvature decreases from the early to middle layers (relative to the first layer of the model), and then increases towards the later layers.

**Strengths:**

- Well written, clear, and concise manuscript
- Investigates a topical question that will be of interest to many in the NeurIPS audience

**Weaknesses:**

W1. Several times throughout the manuscript (including in the abstract and title), it is claimed that the results that larger models that have better next-word-prediction also have less curved trajectories in the early-to-mid layers suggests that models learn straighter trajectories in order to predict better. There is no evidence in this manuscript to support this claim. There is only evidence of correlation between the two, and not of causation. These claims need to be dialed way down and qualified. There can be other causes that lead to both straighter trajectories and better prediction performance. For example, the finding in the later part of the paper that sentence surprisal is correlated with curvature can be exactly this cause: it is possible that the most likely next word is the one that leads to the most "straight" trajectory. Therefore, a model which learns to predict the most likely next word (i.e. a language model) and achieves a good performance, will also have a straight trajectory. The interesting question is why the most likely next word would lead to a straight trajectory, but that is not answered by the current work.

W2. The manuscript heavily leans on a hypothesis developed by previous work (Henaff et al. 2018/9) but it is not clear how the curvature measure defined in the current work is related to the one developed by previous work. This needs to be clarified.

W3. A few possible confounders for the results. See Questions below.

**Questions:**

Major:

Q1. What angles are considered to lead to a "straight" trajectory, and what are the individual values of the curvatures for each layer? Does straight mean closer to 0? Is the curvature considered to increase as arcos increases? As in, the curvature is the highest for arccos = pi when the vectors point in completely opposite directions?

Q2. Currently the average of the curvature between consecutive words/tokens in the sentence is considered, but how does the curvature actually evolve through the sentence? This question also somewhat relates to L127-129: “If the model is achieving next word prediction by reducing the curvature in the trajectory of its internal states over the course of a sentence, then we should observe a reliable decrease in the average curvature across the model layers.” The entailment is not clear to me here. If the model is reducing the curvature over the course of the sentence, then I would expect that there will be a decrease in the curvature that relates to the position of the sentence. It’s not clear why one would expect to see the suggested result across different layers of the network.

Q3. Relatedly, how do the learned positional encodings of GPT-2 interact with the results in this work? Specifically, I am wondering if the reason that the randomly initialized model does not show any substantial changes in curvature is due to the poor representations of position. Perhaps a better baseline would be a model that has pretrained positional encodings but a random initialization of other embeddings and parameters.

Q4. It seems like the surprisal is measured entirely on the test corpus, but it should be tested w.r.t. the training corpus. The models are learning the statistics of language using the training corpus, so one would predict that these statistics are closer to the ones estimated by the models. Perhaps the correlation between surprisal and curvature can be even higher if the surprisal better captures the training statistics.



Minor:
- All the tested models produce token-level embeddings. The curvature computation in Section 2.2 discusses word-level embeddings. Is the curvature computation indeed done on the word-level, and if so, how were the word-level embeddings produced? If the curvature computation is produced by aggregating the token-level embedding within a word using a specific function, such as mean pooling, then how does this function affect the computed curvature? (e.g. when compared to max pooling or taking the last token in the word as the word-level token)

- L6: work from 2019 is hardly recent

- L90 + 6 lines (missing line numbers on page 3): typo “as the difference between to adjacent states“

- Top Fig1, right hand side: words out of order “fox jumps over the” -> “fox jumps the over”

- L234: the earliest citations for the middle layers of language models predicting brain recordings the best is Jain and Huth, 2018 NeurIPS (who show this for LSTM-based models) and Toneva and Wehbe 2019 NeurIPS (who show this for larger transformer and LSTM-based models).


**Limitations:**

Please discuss the fact that there can be other causes for the results you observe (see Weakness 1)

---

> ### Author Rebuttal · Authors · 2023-08-10
>
> W1.
> we agree with the reviewer that the causation is a harder question to answer. And will emphasize in the updated version of the paper that we are showing the evidence at the correlation in this work. We intend to extend our work in causation direction by training models and biasing their representation toward specific curvature values and observe their next word prediction performance. However, we are connecting the next word prediction as behavior of the model to straightening as a hypothesis about evolution of neural trajectories that yield next word prediction behavior. the reviewer points to this fact here. We certainly not claiming that the straightening is the only features that allow the model to develop linguistic competency. As reviewer pointed out we will make this point clear in the discussion.
> W2. The manuscript heavily leans on a hypothesis developed by previous work (Henaff et al. 2018/9) but it is not clear how the curvature measure defined in the current work is related to the one developed by previous work. This needs to be clarified.
> We are sorry about that, our work defines the curvature in the trajectory of representation the in the same manner as henaf 2019, we will clarify this in the text.
> W3. A few possible confounders for the results. See Questions below.
> Questions:
> Major:
>
> Q1.
> We consider straightness with respect to initial layer of the network, we have shown in the supplementary material that a random trajectory on average has a curvature close to 120 degrees and this is indeed the case for untrained and early layers models ( supplementary figure 6A). We considered a trajectory straighter when the average angle of a trajectory decreases with respect to angle in the first layer. in none of the networks we observed curvature to be close to pi
>
> Q2.
> This is a fair point, however here we are measuring an average measure for curvature over the whole sentence, and not at the level of individual words. It is not straightforward to interpret the word-by-word changes in the curvature, one reason for this is that the error in estimating next state in time is cumulative in the sense that the error in predicting the n+1 token will affect n+2 token prediction and so on. As a result, the later part of the sentence could have more variation in the curvature.
>
> Q3.
> This is a good point; we validated that positional information is not contributing to curvature in untrained models.to do so we studied a transformer model with Rotary positional encoding (Su et al. 2021; Biderman et al. 2023). Similar to GPT2 model with learnt positional embedding, we observed that the untrained version of the model does not exhibit any reduction in curvature, supplementary figure 3C,D
>
> Q4.
> This is a great point, we unfortunately don’t have access to the training corpus for many of these models, but agree that it would be a next step. We will try to address this by using models that are trained on publicly available corpus.
> Minor:
> •	All the tested models produce token-level embeddings. The curvature computation in Section 2.2 discusses word-level embeddings. Is the curvature computation indeed done on the word-level, and if so, how were the word-level embeddings produced? If the curvature computation is produced by aggregating the token-level embedding within a word using a specific function, such as mean pooling, then how does this function affect the computed curvature? (e.g. when compared to max pooling or taking the last token in the word as the word-level token)
> For word level embedding we average the activation of tokens that compose the word, and from that point on the curvature computation is identical between the two settings.
> •	L6: work from 2019 is hardly recent
> •	L90 + 6 lines (missing line numbers on page 3): typo “as the difference between to adjacent states“
> •	Top Fig1, right hand side: words out of order “fox jumps over the” -> “fox jumps the over”
> •	L234: the earliest citations for the middle layers of language models predicting brain recordings the best is Jain and Huth, 2018 NeurIPS (who show this for LSTM-based models) and Toneva and Wehbe 2019 NeurIPS (who show this for larger transformer and LSTM-
>
> We thank the reviewer for point these out and we will update the manuscript accordingly

---

> > ### Comment · Reviewer_Sn24 · 2023-08-14
> >
> > Thanks for the response. I agree with the rest of the reviewers that the straightening phenomena is well examined by the authors and that it would be of interest to the NeurIPS audience. I am raising my score to a borderline accept, but I do believe that there is a lot that can be done to strengthen the impact of the paper. For instance, providing more intuition for why straightening is measured the way it is in the work vs other possibilities (e.g. w.r.t. position in a sentence) would be very helpful.

---

> > > ### Author Response · Authors · 2023-08-14
> > >
> > > we thank the reviewer again their insightful comment, and the reconsideration of manuscript. We also agree that we can strengthen the impact of the paper. with regards to measuring straightening, we think this way of measuring curvature is the simplest form, and potentially affect the main effect. for example we could have measure curvature using more parametric approaches ( spline estimation) or over longer temporal window, ( considering multiple words at a time ), but our worry was that it would make the interpretation harder. we will make sure to add points in the the methods, and discussion to clarify our rationale and emphasize reviewers points.

---

### Official Review · Reviewer_S19m · 2023-07-16

**Soundness:** 3 good
**Presentation:** 2 fair
**Contribution:** 3 good
**Rating:** 5
**Confidence:** 3

**Summary:**

This work examines the an hypothesis regarding neural trajectory straightening as a mechanism, by which neural language models achieve next word prediction. Specifically, this hypothesis connects between the objective of next word prediction and extrapolation to the embedding of the next word in neural representation space. They define layer curvature based on prior work and find that (1) autoregressive LMs consistently reduced their curvature from early to middle layers, and this effect was only observed for trained models; ,odel size and training dataset size affected the model’s ability to reduce curvature; model-generated sentences exhibited lower curvature compared to natural human-generated sentences; average curvature correlated with average sentence surprisal in the middle layers of the model.


**Strengths:**

This paper explores an interesting hypothesis regarding the connection between an internal geometric property of Transformer based LMs and their performance. If this was not examined in prior work, I find the questions posed in all 4 experiments novel and interesting, and the results non-trivial. In particular, I liked the thoroughness in testing both different model sizes and different training set sizes for the same model size, that did a good job in removing an important confounder in my opinion (though not enough discussion and experimentation regarding the point in 1B tokens training set that broke the trend)


**Weaknesses:**

I find that this paper can be strengthened from several different angles:

- Discussion of prior work: I am not familiar with literature on geometric interpretability of language models. This paper is on this exact topic but does not convey sufficient background on related work.
- Question scope: the focus on one specific geometric measure limits this paper’s strength, to me it seems that several related measures can be examined. Alternatively, though it’s intuitive, I find the focus on this specific geometric measure as not sufficiently motivated.
- Depth of investigation: For each experiment, only the basic setup was ran and often there was not sufficient discussion on the outcome or follow up experimentation (eg, what happens in the second half of the network? Why did the 1B token experiment not show the same trend as 1M, 10M, 100M?)
- Several experimental design choices were not sufficiently motivated (Why only one dataset of 8,408 sentences? Why constrain sentences to be between 6 and 19 words long, and to not contain abbreviations or uncommon words?)
- writing and presentation. There were several clumsy sentence phrasings (eg, first intro sentence) and some typos (eg, mid sentence capitalization line 121). More importantly, some core quantities were not adequately presented (eg, no formula given for the employed 3-gram surprisal metric), and the figures were generally pretty hard to decipher (eg, What does figure 4A mean? I didn’t understand what quantity is referred to by the title: “The predictions of the representation straightening hypothesis”. What do the axes of this plot correspond to?). As mentioned above, I found the results sections not written well enough, often reiterating the premise and the intuition and not conveying and discussing the actual experimental outcome clearly enough.

**Questions:**

- Why only train and not compare to other models?
- The y-axis in your reported plots is "curvature change" What was the distribution of the original absolute curvature angle in the first layer? If you report only a relative number, how can I know that curvature drop makes it close to zero  doesn’t end up in a negative degree, rendering the conclusion on "straightening" incorrect? To be clear I'm pretty sure / hopeful that the authors will have a good answer and that "straightening" indeed takes place, but this represents some weakness in the presentation.
- I found this specific sentence very odd: "The results generalize to smaller models and other surprisal metrics (other n-gram metrics and PCFG-parser-based surprisal - not shown here)" why do the authors mention other related results but do not show them?
- Why did you use n-gram surprisal and not LM perplexity in the forth experiment? The first is model agnostic and the latter is model related, so maybe you should have ran both, but I find perplexity very natural to use here.

**Limitations:**

Yes.

---

> ### Author Rebuttal · Authors · 2023-08-10
>
> Discussion of prior work:
> Thanks for the reviewer for bringing this issue to our attention, we will include a more through discussion of prior work in our revised manuscript. There are a number of prior work that utilized geometric approaches to understand the internal representation of language models. (Hewitt and Manning 2019) used linear transformation to identify projection of over model representation that is maximizing similarity to syntactic distances in a parse tree, and found that middle layers of BERT provide best representation.In another line of work (Mamou et al. 2020) used manifold analysis to uncover how different features of words and sentences, such as part of speech ) become separable in deep layers of language models like BERT, and GPT. More recently, (Valeriani et al. 2023) have investigated how the geometric properties such as intrinsic dimension changes across the layers of transformer models, finding that intrinsic dimension first increases before sharply decreeing in deeper layer of the bidirectional transformer models. While these works provide insight to geometrical properties of these network, to our knowledge no prior work have tested a representational level hypothesis that connect behavior of the network (next word prediction) with neural trajectories of individual sentences.
>
> Scope:
> The reviewer brings up a good point, and we will include more information regarding our motivation to focus on straightening. We specially focused on straightening because it provides a testable prediction about how model behavior is shaped by its internal representation. Figure 4 is one such test in which we used our geometric measure to reveal a possible mechanism leads model to deviate from natural language, mainly that their generated sequences exhibit more straightening compared to human produced language.
> Depth of investigation:
>
> We agree with the reviewer that there are aspects of the results that we did not discuss thoroughly. We hope to investigate these phenomena in future work, and tried to address some of the follow up experiments in the supplementary section.
>
> experimental design :
> We thank the reviewer for bringing these to our attention, and will fix the errors and clarify other topics in the updated version of the paper.
> 1.	Choice of dataset: we used universal dependencies because we wanted to be able to quantify the sentences and phrases along multiple lexical, semantic and syntactic feature. UD corpus provides precomputed values for syntactic features. We further restricted our sentences to include 100K most common words in English (using google ngram dataset). Our choice of sentence length was to ensure we have enough sensitivity in measuring average curvature. Sentences that are very long would result in averaging out the differences that we are interested in.
> 2.	We will modify figure 4A to emphasize that it is for illustrative purposes and will clarify the detail. the axes in this figure correspond to state spaces built by units in layer P of the network. The trajectories represent evolution of unit activity when the model is generating a sentence vs when it is exposed to a veridical sentence. If the unit activity is favoring a straight trajectory, then the model-generated sentences would be have lower curvature (red compared to blue). Figure 4.B shows the results of an experiment in which we provided a model with 3 context tokens and recorded its generated sequence for 7 tokens, along with the model representation for the generated sequences, and contrasted it with a condition we provided the full 10 token sequence to the model. We then compare the curvature for generated to the ground-truth, and observed that model generated sequences achieve lower curvatures, in line with our prediction in figure 4.A.
>
> Questions:
> Why only train and not compare to other models?
> we added new models in supplementary figure 3 (RWKA, GPT-NEOX, OPT,GPT2-XL). We observed similar decrease in the curvature in trained version of model, as well as similar correlation between curvature and surprisal.
> The y-axis in your reported plots is "curvature change" What was the distribution of the original absolute curvature angle in the first layer? If you report only a relative number, how can I know that curvature drop makes it close to zero doesn’t end up in a negative degree, rendering the conclusion on "straightening" incorrect? To be clear I'm pretty sure / hopeful that the authors will have a good answer and that "straightening" indeed takes place, but this represents some weakness in the presentation.
>
> We focused on curvature change so that we can compare models in their performance for curvature reduction. However, we agree that the absolute curvature values are important to consider as well. To this end we added supplementary figure 6 to clarify this. First using simulation, we showed that for a trajectory over random points in space, the average curvature is distributed around 120 degrees. For many of the models we tested, early layers indeed have a curvature in the range of 120, and there is a drop in the curvature in deep layers of the network.
> I found this specific sentence very odd: ...
>
> We are sorry for not including this in the main part of manuscript. We included the additional surprisal metric as well as the relationship between surprisal and curvature for a number of new models in the supplementary information supplementary figure 4.
> Why did you use n-gram surprisal and not LM perplexity in the forth experiment? The first is model agnostic and the latter is model related, so maybe you should have ran both, but I find perplexity very natural to use here.
>
> We picked a model agnostic measure so we could test it across many models as the reviewer mentioned. But we agree and included a new figure in the supplementary that is relating curvature to LM perplexity. We included the results for the main model (GPT-XL) and few other models in supplementary figure 3.

---

> > ### Comment · Reviewer_S19m · 2023-08-13
> >
> > I thank the authors for addressing many of my comments and questions. Given their responses, I am raising my score.

---

> > > ### Author Response · Authors · 2023-08-13
> > >
> > > We are glad to hear that we were able to address reviewer's questions, and appreciate their reconsideration of the work.

---

### Official Review · Reviewer_ycZy · 2023-07-27

**Soundness:** 3 good
**Presentation:** 3 good
**Contribution:** 3 good
**Rating:** 6
**Confidence:** 3

**Summary:**

This paper provides evidence that autoregressive language models - specifically the GPT family straighten the internal trajectory of word sequences, making them more linear, in order to better predict next words. They show that trained models decrease sequence curvature across layers, larger models straighten more, model-generated sentences are straighter, and curvature correlates with unpredictability.

**Strengths:**

The paper introduces computational evidence for the trajectory straightening hypothesis using a simple and intuitive curvature metric and backs it up with various experiments. It is also well-written and raises exciting questions about the working/interpretability of these models.

**Weaknesses:**

The paper did not perform any ablation studies to see what is causing the trajectory straightening. Removing different components of the transformer like the feed-forward layer, and seeing the effects on straightening may lead to some more insights. It is not clear if straightening depends on the transformer architecture specifically or also occurs in other model architectures like LSTMs or vanilla RNNs Do similar dynamics occur in MLPs?

**Questions:**

Does straightening occur in LSTM or GRU sequential models? Or is it unique to transformer self-attention?
If we remove layers or transformer components like self-attention or the feed forward layer is straightening impeded?
Does straightening also happen in other transformer-based models but with a different pertaining objective such as BERT ?


**Limitations:**

The authors have a sufficient limitations section.

---

> ### Author Rebuttal · Authors · 2023-08-10
>
> Weaknesses:
> The paper did not perform any ablation studies to see what is causing the trajectory straightening. Removing different components of the transformer like the feed-forward layer, and seeing the effects on straightening may lead to some more insights. It is not clear if straightening depends on the transformer architecture specifically or also occurs in other model architectures like LSTMs or vanilla RNNs Do similar dynamics occur in MLPs?
>
> We appreciate the reviewer for pointing this out. To address this, we performed ablation on GPT2-XL autoregressive model (Supplementary figure 1). We performed ablation at different layers of the network [5,15,25,35,45], and for each layer on individual modules [attention head, attention projection, MLP]. To do the ablation in each module we set the weights with identity matrix and biases to 1. Specifically, for attention head we replace the weight for Key matrix to identity, for attention projection we replace the weights with identity, and for MLP we replace the two weight matrices (h*4h, 4h*h, where h=hidden size=1600) such that the effective weight is identity. This approach allowed us to do minimal disruption in on model and observe how the representation of ablated layer is transformed.  For computational considerations, in each experiment we tested 500 randomly sentences sampled uniformly.
> We observed 2 main effects. First, ablation of early layers has more consequence on curvature in the succeeding layers. Second, we observed that the ablation of attention mechanism has leads to largest deficit in reducing curvature in the succeeding layers. This suggest that attention mechanism is causing the straightening.
> We also tested recent transformer like RNN model ((Peng et al. 2023)) and observed similar straightening to transformers (Supplementary figure 3).
>
> Questions:
> Does straightening occur in LSTM or GRU sequential models? Or is it unique to transformer self-attention? If we remove layers or transformer components like self-attention or the feed forward layer is straightening impeded? Does straightening also happen in other transformer-based models but with a different pertaining objective such as BERT?
>
> We thank reviewer for bringing this issue up. This is a very important question that we intend to address in our continuation of this work. Our current results points towards objective function as the main driver of straightening. As reviewer suggested, we tested the straightening hypothesis in a bidirectional model ( Bert-large-uncased, Supplementary figure 2).  We observed that a different pattern of curvature across layers. Early layers do not show a gradual drop in curvature, something that we observed in autoregressive transformer models. instead, there was a drop in curvature in deeper layers. Importantly there is no reliable relationship between surprisal and curvature for across the layers. What could contribute to curvature reduction in deeper layers of BERT. It is possible that a masked language modeling objective still share some similarities to an autoregressive objective. For example, cases masked words that are towards late part of a sequence of tokens, the model can use information from past words to predict missing masked token.
>
> Limitations:
> The authors have a sufficient limitations section.

---

> > ### Comment · Reviewer_ycZy · 2023-08-18
> >
> > Thank you authors for your diligent response. I agree with the other reviewers that the straightening phenomena is interesting and would be of interest to the wider NeurIPS audience. However, the paper does need some work to be improved in terms of writing and explaining the results. I would recommend adding the result on "objective function as the main driver of straightening" with experiments showcasing the BERT results in the main paper. Thus, I would like to keep my original score.

---

> > > ### Author Response · Authors · 2023-08-21
> > >
> > > we appreciate reviewers insightful inputs. We will indeed follow their suggestion to include a new section on "objective function as the main driver of straightening" in which we will discuss the results for BERT and emphasize how in autoregressive models, a predictive objective function can lead into straightening of neural trajectories, and connect it to a mechanistic description of model behavior.

---

### Author Rebuttal · Authors · 2023-08-10

---

> ### Comment · Reviewer_Sn24 · 2023-08-14
>
> The submitted rebuttal PDF is 6 pages. I believe the allowed length was 1 page. I will not look at anything past the 1st page as it is not fair to allow this work to provide more information than other works.

---

> > ### Author Response · Authors · 2023-08-14
> >
> > we are sorry for the length of pdf document. it only contains figures, and for clarity we enlarged the figures, and put them in seperate section and this resulted in increased number of pages. If the reviewer would like we can reorganize the figures and plots so that we reduce the number of pages.

---

### Decision · Program_Chairs · 2023-09-21

**Decision:**

Accept (poster)

**Comment:**

The paper examines whether transformers conduct “trajectory straightening”. Straighter trajectories, it is argued, should facilitate generalization. The work proposes an easy to understand curvature metric and conducts several experiments using it, showing that indeed, in later layers the layer representations become straighter. The reviewers are leaning positive and the rebuttal alleviated some concerns. I think this is an interesting finding with interesting connections to previous work, and it opens up further study into how different architectures behave, how language models generalize, etc.

I strongly encourage the authors to revise the manuscript to incorporate the reviewers' suggestions. In particular, I recommend that the claims around trajectory straightening be dialed way down and qualified — it would be best to simply state it as a hypothesis without assuming any causation, and then conducting experiments to examine the hypothesis.